# The Performance of Representative Asian Vegetables in Different Production Systems in Texas

Genhua Niu [1,*], Joseph Masabni [1,*], Triston Hooks [1], Daniel Leskovar [2] and John Jifon [3]

1 Texas A&M AgriLife Research and Extension Center at Dallas, 17360 Coit Road, Dallas, TX 75252, USA; tnhooks@gmail.com
2 Texas A&M AgriLife Research and Extension Center at Uvalde, 1619 Garner Field Road, Uvalde, TX 78801, USA; daniel.leskovar@ag.tamu.edu
3 Texas A&M AgriLife Research and Extension Center at Weslaco, 2415 E Hwy 83, Weslaco, TX 78596, USA; john.jifon@ag.tamu.edu
* Correspondence: genhua.niu@ag.tamu.edu (G.N.); joe.masabni@ag.tamu.edu (J.M.)

**Abstract:** Demand for Asian vegetables is rising rapidly due to changing demographics and increasing consumer awareness of their health benefits. However, growers are not familiar with growing these "foreign" crops due to insufficient technical information regarding suitable cultivars for different regions, production schedules, disease and pest susceptibility, and postharvest management. The objective of this study was to conduct trials in different production systems and climate regions to demonstrate the potential of growing Asian vegetables in Texas. We conducted preliminary trials of nine leafy greens in the open field, high tunnel, and greenhouse (container and hydroponic production) to explore the suitability and potential for year-round production. We also conducted field trials for warm season crops in the open field in different climate zones. Results indicated that for cool season leafy greens, open field production has a limited growing season, high tunnel has the potential to extend the growing season, while greenhouse may provide year-round production using soilless substrate container culture or hydroponic system. For warm season crops, early planting is recommended for high yield. Additional research is warranted in different regions to test more species and cultivars and optimize the production system of high-performing cultivars to maximize production and profitability.

**Keywords:** Bok choy; eggplant; mizuna; profitability; seasonal production; tatsoi; yardlong bean

## 1. Introduction

Asian vegetables are defined as vegetables that are prominent in Asian cuisine but are relatively less well-known in the Western diet. Although these crops have been domesticated and cultivated for millennia, their introduction into the Western cuisine has been slow [1,2]. Yet, Asian vegetables are becoming more popular around the world due to changing demographics and evolving consumer taste with increased interest in their distinctive flavor and texture as well as increased awareness of their rich nutritional quality [2,3]. In a report by the Australian Rural Industrial Research and Development Corporation [4], more than 80 Asian vegetable types are produced in Australia. However, most in-demand types of Asian vegetables are limited in number and only about 16% of their produce is exported. In Europe, although the volume of fresh Asian vegetable consumption is slowly increasing, the outlook is positive due to increasing demand for diverse, healthy, and exotic vegetables. However, there have been very few studies conducted on Asian vegetables in Europe [5].

In the United States, Asian and other ethnic vegetables are gaining popularity, not only among ethnic groups, but also among non-ethnic groups because of the expansion of a diverse diet, increased interest in Asian cuisine, awareness of healthy eating, and easy access to recipes [6,7]. In recent years, the increasing demand for ethnic/Asian

vegetables has resulted in a rapid expansion of these vegetables, and more than 40 types are commercially grown in Florida [8–10]. Approximately 70 types of Asian vegetables are produced in the Central Valley of California [11]. Similar to other vegetables, Asian vegetables produced in California are shipped to all other states in the U.S. According to the USDA report on Annual Fresh Fruit and Vegetable Shipments (2015), domestically grown Asian vegetables make up 43% of the total harvest. Most of the imported produce comes from Mexico, with a small amount from South Korea and China. However, long-distance transportation of many Asian vegetables results in poor quality when delivered to grocery stores [12].

Texas ranks third in the nation in Asian population. The rapid expansion of the Asian population in Texas presents significant opportunities for producers, especially those near densely populated areas such as Dallas, Houston, Austin, and San Antonio. Therefore, Texas has great market potential and production opportunities for Asian vegetables. Compared to northern states, Texas' climate has advantages in growing many leafy Asian vegetables, which are mostly cool-season vegetables and can be grown as fall, winter, and/or spring crops. However, high temperature spikes in fall and spring can negatively impact the growth of young seedlings and can induce bolting in cool-season leafy greens such as Bok choy (*Brassica rapa* subsp. *chinensis* L.). In addition to temperature, daylength may also lead to bolting, especially for species sensitive to photoperiod. By using high tunnels or greenhouses, the growing season can be extended further in order to avoid high or low temperatures. Additionally, there is little information on disease and pest management and on best-suited varieties for different regions and production systems.

Bok choy is one of the most popular Asian leafy vegetables. Bok choy grows best at 18 °C to 20 °C, but it can tolerate temperatures as high as 35 °C and as low as −3 °C [13]. It is widely grown in subtropical and temperate regions year-round and is increasingly expanding in Florida and other states in recent years because of its great profitability (high yield and high popularity) [13,14]. In Texas, Bok choy is also one of the popular Asian leafy greens produced by small-acreage vegetable farmers both conventionally and organically (personal communication with Texas Organic Farmers and Gardeners Association). Among warm Asian vegetables, eggplant (*Solanum melongena*) is one of the popular vegetables based on its availability by most major (non-Asian) seed companies. Its long, slender, and thin skin are the main characteristics differing from the commonly sold globe shape type [15,16]. Another representative crop is yardlong bean (*Vigna unguiculata* ssp. *sesquipedalis*), a leguminous vegetable crop with climbing vines that produces long pods consumed as a cooked vegetable [17,18]. It is also called asparagus bean and is related to cowpea (*Phaseolus vulgaris*).

To successfully introduce these popular Asian vegetables to Texas, it is necessary to better understand their adaptability and suitability. Therefore, the objective of this study is to conduct trials in different production systems, locations, and growing seasons to better understand these concerns.

## 2. Materials and Methods

### 2.1. Leafy Vegetable Trials in Greenhouse Containers in West Texas

Nine cool-season leafy vegetable seeds were purchased from Kitazawa Seed Company (Table 1). Two greenhouse trials were conducted in El Paso, TX (31.7619° N, 106.4850° W) from October to December 2018. Seeds were sown on 17 October (Expt. 1) and 1 November (Expt. 2) 2018 in 72-cell trays filled with Metro-mix 360 (SunGro Hort., Bellevue, WA, USA) and placed on heating propagation mats on a mist bench in the greenhouse. After germination, seedlings were moved out from the mist and put on benches in a greenhouse. On 31 October (Expt. 1) and 14 November (Expt. 2) 2018, uniform seedlings were transplanted into 15.24 cm containers (1.8 L) and arranged in a completely randomized design with 10 replications per cultivar. Plants were irrigated with 500 mL of nutrient solution whenever the surface substrate was dry. The nutrient solution was prepared by adding 1.0 g L$^{-1}$ (150 mg L$^{-1}$ N) 15N-2.2P-12.5K (Peters 15-5-15 Ca-Mg



Special; Scotts, Marysville, OH, USA) to reverse osmosis (RO) water. The greenhouse was equipped with a natural gas heating system. The average temperature, daily light integral (DLI), and relative humidity were $25.10 \pm 1.51$ °C (average $\pm$ standard deviation), $14.03 \pm 2.43$ mol m$^{-2}$ d$^{-1}$, and $29.61 \pm 5.88\%$, respectively, for the first experiment, and $24.65 \pm 1.29$ °C, $11.51 \pm 3.15$ mol m$^{-2}$ d$^{-1}$, and $28.17 \pm 5.47\%$, respectively, for the second experiment.

**Table 1.** List of cool-season leafy vegetables used in greenhouse trials in El Paso, Texas.

| Cultivar Name | Scientific Name |
| --- | --- |
| Beka Santoh | *Brassica rapa*-Pekinensis group |
| Chinese | *Brassica rapa*-Chinensis group |
| Ching Chang | *Brassica rapa* |
| Green Wave | *Brassica juncea* |
| Petite Star | *Brassica rapa*-Chinensis group |
| Purple Magic | *Brassica rapa*-Chinensis group |
| Red Mizuna | *Brassica juncea* |
| Tatsoi | *Brassica rapa*-Narinosa group |
| Tatsoi Savoy | *Brassica rapa*-Narinosa group |

Seeds of the above list of vegetables were purchased from Kitazawa seed company (www.kitazawaseed.com) on 10 September 2018.

Plants were monitored for pests and diseases. On 26 November (Expt. 1) and 10 December (Expt. 2) 2018, all plants were harvested, and visual quality was rated. Visual quality was scored using a scale of 1 to 5, where 1 = complete plant death, 2 = severe stress symptom, 3 = moderate stress symptom, 4 = some stress symptom, and 5 = no stress symptom. Increments of 0.5 were used to improve the assessment for plants that were difficult to score. Shoot fresh weight (yield) was recorded. Leaf Brix (soluble sugar content) was measured using a refractometer (Extech Instruments, Nashua, NH). Juice from leaves was extracted using a garlic press and Brix (%) was recorded.

### 2.2. Leafy Vegetable Trials in Greenhouse Hydroponic Systems in West Texas

Two greenhouse trials were conducted in different hydroponic systems from April to July 2019 in El Paso, Texas. Seeds of the eight cultivars (Table 1) excluding Tatsoi were sown in rockwool cubes on 05 April 2019 (Expt. 1). The rockwool cubes were thoroughly rinsed with distilled water until the electrical conductivity (EC) reached 1.2 dS m$^{-1}$ with a pH of 6.2. On 22 April 2019, uniform seedlings were transplanted to different production systems. Hoagland nutrient solution was prepared by adding fertilizers to RO water. The nutrient element concentrations in mg L$^{-1}$ were 210 (nitrogen), 31 (phosphorus), 234 (potassium), 200.5 (calcium), 48.6 (magnesium), 64 (sulfate), 0.5 (boron), 0.5 (manganese), 0.05 (zinc), 0.02 (copper), 0.01 (molybdenum), and 5.97 (chelated iron). On 13 May 2019, plants were harvested, and fresh weight of shoots was recorded. For the second experiment, seeds were sown on 3 June 2019, transplanted on 12 June, and harvested on 3 July 2019. The same protocols for seed germination and culture methods were followed for the two experiments. To prevent aphids and other insects, all plants were sprayed weekly with insecticidal soap in the early morning before 8 a.m. The greenhouse was equipped with a pad-and-fan cooling system and a shade cloth with 30% light exclusion was used on top of the roof. The average temperature, DLI, and relative humidity in the first experiment were $23.71 \pm 0.66$ °C, $13.43 \pm 1.90$ mol m$^{-2}$ d$^{-1}$, and $42.93 \pm 6.21\%$, respectively, and for the second experiment, $27.16 \pm 1.35$ °C, $15.43 \pm 2.92$ mol m$^{-2}$ d$^{-1}$, and $41.90 \pm 6.10\%$, respectively.

The three hydroponic systems used in this trial were nutrient film technique (NFT), deep water culture (DWC) (Figure 1), and Ebb and Flow system (EbbFlow). Each NFT unit (GT50-612, FarmTek, Dyersville, IA, USA) consisted of four troughs that measured $200 \times 10 \times 5$ cm with a 3% slope. Each trough had 12 slots for a total capacity of 48 plants per NFT. A reservoir tank (150 L capacity; Premium Reservoir, Botanicare, Chandler, AZ, USA) and pump were connected to each NFT unit and provided continuous recirculation of nutrient solution to the troughs. The DWC system is a Botanicare bin, placed on top of

the greenhouse bench, and measured 0.6 m wide × 1.2 m long × 0.2 m deep. A floating styrofoam board (0.6 m × 1.2 m) with 40 holes (4.0 cm in diameter) evenly distributed across the board was used to support the net pots (5.0 cm in diameter) where plants were placed. For the EbbFlow system, the same Botanicare bin was used without a floating board and was placed on top of the bench. Seedlings were transplanted into 10.5 cm square pots (500 mL) filled with perlite and vermiculite at an equal volume ratio. A total of 32 plants were placed in the bottom of the bin. In addition, a nutrient reservoir (100 L) was connected to the bin and a submersible pump was used to provide the nutrient solution to the bin four times daily (15 min each), which was controlled by a timer. There were three replications (units) in each hydroponic system. The experiments followed a split-plot design with a hydroponic system as the main plot and cultivars sub-plot. The same nutrient solution (Hoagland formula) mentioned above was used for all systems. For nutrient replenishment, a half-strength solution was used to keep the reservoir or solution bin at the same level.

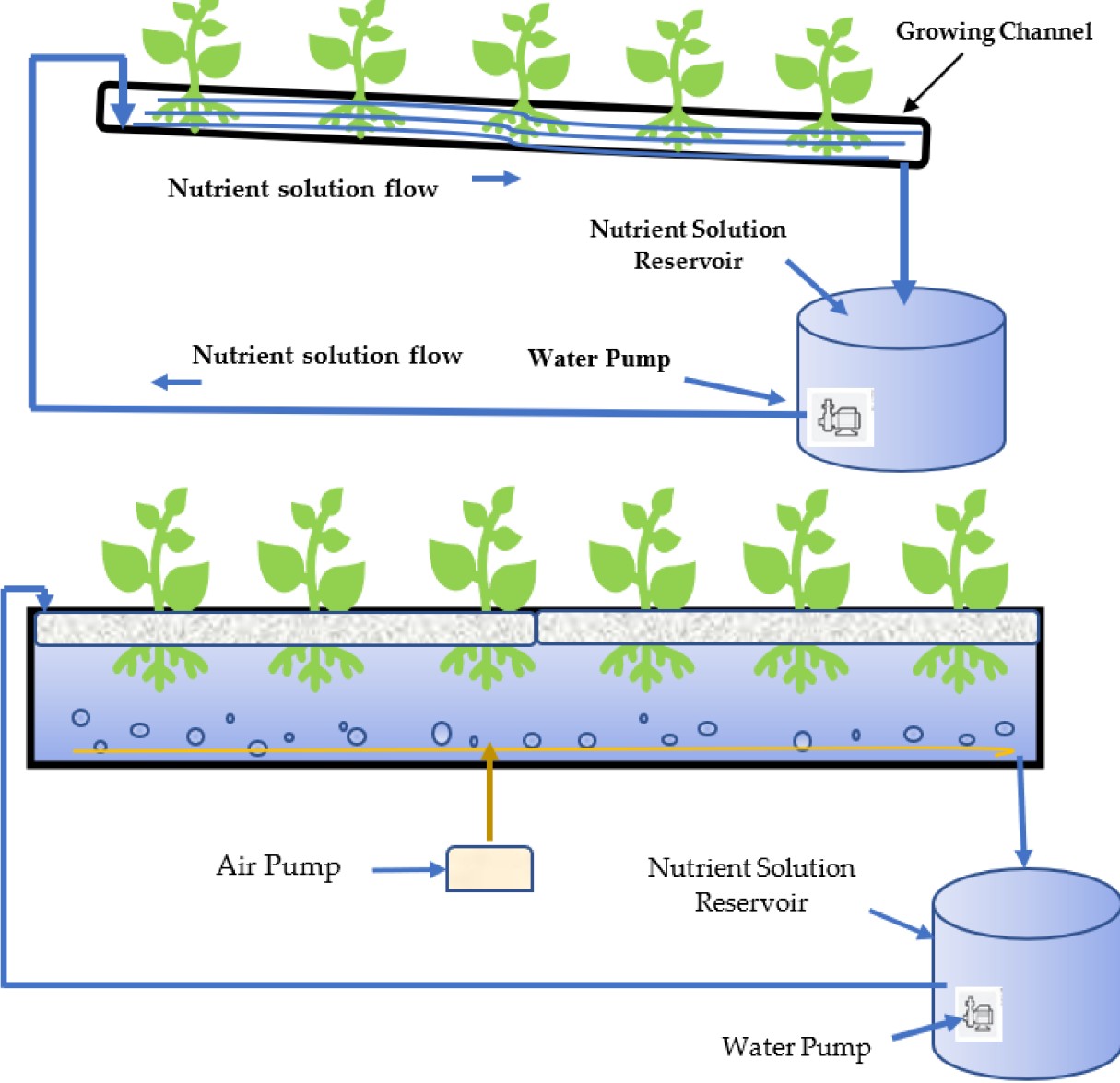

**Figure 1.** Schematic diagram of Nutrient Film Technique (NFT, **top**) and deep-water culture (DWC, **bottom**) hydroponic systems.

*2.3. Leafy Vegetable Trials in the Open Field and High Tunnel in East Texas*

In East Texas, three trials of cool-season leafy vegetables were conducted between September 2018 and February 2019 at the East Farm of the Research and Extension Center at Overton, Texas (32.2746° N, 94.9786° W). Trials 1 and 2 were conducted in an open field on plastic mulch in rows spaced at 1.8 m x 2.2 m, while the third was conducted in a high tunnel (6 m × 18 m). For all trials, plants were planted in plastic mulch rows spaced at 0.9 m × 1.6 m. The nine varieties in Table 1 were used in these trials. The field soil was tested with the following results: pH of 7.0 and nutrient element concentrations (mg L$^{-1}$) were nitrate-N, 12; phosphorus, 21; potassium, 121; calcium, 892; magnesium, 111, and sulfur, 5. Thus, fertilizers were applied (see below). For high tunnel soil, the pH was 7.4, and nutrient concentrations were (mg L$^{-1}$): nitrate-N, 151; phosphorus, 78; potassium 665; calcium 2503; magnesium, 360; and sulfur, 384. Thus, no additional fertilizers were needed.

Seeds were sown on 17 September, 8 October, and 12 October 2018, for trials 1 and 2 (field) and 3 (high tunnel), respectively. A similar methodology was adopted as those in the El Paso trials for the germination of seeds. For trial 1, plants were transplanted to the field on 4 October with 28 plants per plot with three replications and harvested on 16 November (43 days after transplanting). Plants were fertilized using 11N-25P-3.3K (Soludrip 11-56-4 by Vital Fertilizers, Mission, TX, USA) which was incorporated in the field during bed preparation at a rate of 6.2 g m$^{-2}$. The pre-plant fertilizer application was not applied for trial 2 as the beds were already made and were used for a prior trial. The nutrient solution was prepared by adding 0.3 g L$^{-1}$ (48 mg L$^{-1}$ N) 16N-7.0P-13.3K (Soludrip 16-16-16) and was applied twice to all three trials at 2-week intervals after planting. For trial 2, plants were transplanted in the field on 6 November 2018. Due to cold winter weather, plants were growing very poorly, and the plants were eventually harvested on 8 February of the following year (94 days after transplanting). For trial 3 (high tunnel), 16 plants per plot (0.9 m × 1.6 m) were planted with 3 replications (plot). The end doors and side curtains of the high tunnel were not closed, that is, the high tunnel was just covered with a plastic film on the roof. The plants were transplanted to the high tunnel on 03 November 2018 and harvested on 14 December 2018 (43 days after planting). For all trials, plants (plots) of different cultivars were arranged in a randomized complete block design with three replications. Data collection included visual rating, shoot fresh weight (yield), and leaf Brix. Leaf Brix was measured using the same methodology described above for El Paso experiments. For plant rating, a different scale was used: where 1 = excellent stand, and plants are uniform in height, 2 = little variation in height, 3–4 = more variation in plant height, 5 = average plant height and uniformity, 8–9 = significant variation in plant height and/or thin stand, and 10 = dead plants. For all trials, plants were monitored for insects and diseases, and Sevin insecticide was used only once in trial 1.

*2.4. Warm-Season Vegetable Trials in West Texas*

Two warm-season crops (Table 2) were used in this trial in El Paso, Texas. The climate in this location is a typical semi-arid climate with an average annual rainfall of 200 mm. The average temperature during the trial period ranged from 21 °C to 33 °C with very low relative humidity (10 to 30%) especially before the monsoon season, which typically starts in early July. Seeds of the three cultivars of eggplant and yardlong bean were purchased from Kitazawa Seed Company. Seeds were sown on 15 March 2019, in 72-cell trays filled with Metro-mix 360 and placed in a mist bench in the greenhouse. Germinated yardlong bean seedlings were transplanted on 2 April into square plastic pots (500 mL) with Metro-mix 360, while eggplant seedlings were transplanted on 8 April. Yardlong bean and eggplant were transplanted to the outdoor raised bed (1.5 × 6 × 0.2 m) on 23 April and 16 May, respectively. A trellis was constructed using T-posts for the yardlong bean.

**Table 2.** List of cultivars of eggplants and yardlong beans used in the field trials in El Paso in west Texas and Overton in east Texas.

| Crop | Scientific Name | Cultivar Name |
| --- | --- | --- |
| Yardlong Bean | *Vigna unguiculata* ssp. *sesquipedalis* | Akasanjaku |
| Yardlong Bean | *Vigna unguiculata* ssp. *sesquipedalis* | Kurosanjaku |
| Yardlong Bean | *Vigna unguiculata* ssp. *sesquipedalis* | Yu Long |
| Eggplant | *Solanum melongena* | Millionaire |
| Eggplant | *Solanum melongena* | Purple Shine |
| Eggplant | *Solanum melongena* | Shoya |

Seeds of the above list of vegetables were purchased from Kitazawa seed company (www.kitazawaseed.com) on 4 February 2019.

The raised beds were covered with garden fabric to prevent weed growth. Plants were irrigated with drip irrigation twice daily in the morning and afternoon. Slow-release fertilizer (Osmocote 15-9-12) was applied at 10 g per plant after transplanting. A sulfur amendment was also applied at 1 L per plant due to high initial pH (8.3) based on initial soil analysis (mg L$^{-1}$: nitrate -N, 64; phosphorus, 383; potassium, 867; calcium, 6200; magnesium, 514; and sulfur, 107). In addition, water-soluble nutrient solutions at 200 mg L$^{-1}$ N (Peter's 15-5-15) were applied weekly. Fruits were harvested twice weekly starting in early June (yardlong bean) or middle June (eggplant) and terminated in early August, and cumulative yield was reported.

*2.5. Warm-Season Vegetable Trials in East Texas*

The same warm-season cultivars in Table 2 were used in the east Texas field trial. Each cultivar had four replications (plots) organized in a randomized complete block design. Yardlong bean was planted at 0.33 m apart and eggplant at 0.66 m apart within rows that were spaced 1.7 m center to center. The field was plowed, and disced and plastic mulch was laid on 17 May 2019. All cultivars were planted on 23 May 2019. A trellis, consisting of T-posts spaced 3 m apart and cattle panels were used for the yardlong bean. Carbaryl (Sevin) insecticide was sprayed on 28 June and 5 July on all cultivars to control insects, and Ecotec (OMRI approved broad spectrum miticide containing rosemary and peppermint oil), was sprayed on 23 July for spider mite control in eggplants. Yardlong bean was harvested 13 times between 26 June and 29 July. Eggplant was harvested five times between 9 July and 29 July 2019.

**3. Results and Discussion**

*3.1. Leafy Vegetable Trials in Greenhouse Container Culture*

All nine cultivars germinated quickly (within 3 days) and uniformly and performed well in container culture. 'Chinese' had the highest yield (shoot fresh weight), followed by 'Beka Santoh' and 'Ching Chang' in Expt. 1 and all others had similar shoot fresh weight, which was about half that of 'Chinese' or less (Table 3). In Expt. 2, 'Beka Santoh' and 'Chinese' had the highest fresh weight, followed by 'Ching Chang'. 'Purple Magic' had the lowest. The yield results of Expt. 2 were slightly lower, but trends were generally consistent with those in Expt. 1. The lower yield in Expt. 2 was partially due to a lower DLI and insect damage, while the average daily temperature was only a half-degree lower. In both experiments, thrips and aphids were the two pests affecting most cultivars. 'Green Wave' and 'Red Mizuna' did not have thrips or other pest damage, while all others did. In addition to thrips, most cultivars of *B. rapa* were infested with aphids. While the quantitative rating of the insect damage was not rated, there were visible differences among cultivars. Also, plants in Expt. 2 had relatively more pest damage compared to those in the first experiment, especially 'Ching Chang'. We did not spray any chemicals during these studies.

**Table 3.** Yield (fresh shoot weight), leaf Brix, and visual scores of leafy vegetables grown in containers in a greenhouse in El Paso, TX, USA.

| Cultivar | Fresh Weight (g/Plant) | | Brix (%) | | Visual Score | |
|---|---|---|---|---|---|---|
| EXPT 1 (31 October–26 November 2018) | | | | | | |
| Beka Santoh | 265 | B $^z$ | 2.92 | B | 4.84 | ABC |
| Chinese | 308 | A | 4.12 | A | 4.76 | ABC |
| Ching Chang | 249 | B | 2.52 | BC | 4.64 | BC |
| Green Wave | 144 | C | 4.72 | A | 4.92 | A |
| Petite Star | 136 | C | 2.00 | CD | 4.68 | ABC |
| Purple Magic | 72 | C | 1.84 | D | 4.60 | C |
| Red Mizuna | 118 | C | 3.00 | B | 4.88 | AB |
| Tatsoi | 148 | C | 2.56 | BC | 4.78 | ABC |
| Tatsoi Savoy | 159 | C | 4.16 | A | 4.82 | ABC |
| EXPT 2 (14 November–10 December 2018) | | | | | | |
| Beka Santoh | 236 | A | 4.52 | AB | 4.72 | ABCD |
| Chinese | 226 | A | 4.32 | AB | 4.74 | ABC |
| Ching Chang | 146 | B | 5.00 | AB | 4.36 | E |
| Green Wave | 98 | D | 3.64 | B | 4.74 | ABC |
| Petite Star | 100 | D | 4.00 | B | 4.78 | AB |
| Purple Magic | 60 | E | 3.72 | B | 4.78 | AB |
| Red Mizuna | 83 | DE | 4.32 | AB | 4.88 | A |
| Tatsoi | 139 | BC | 5.04 | AB | 4.58 | CD |
| Tatsoi Savoy | 112 | CD | 5.52 | A | 4.56 | D |

$^z$ Means followed by different letters within columns indicate significant variety differences according to Tukey's honestly significant difference test ($p < 0.05$).

For leaf Brix, 'Chinese', 'Green Wave', and 'Tatsoi Savoy' had higher values, all above 4, while 'Purple Magic' had the lowest value of 1.84. The Brix values in Expt. 2 were slightly higher than those in Expt. 1, possibly due to slight changes in growing conditions. 'Tatsoi Savoy' had the highest Brix (5.52), 'Green Wave', 'Petite Star', and 'Purple Magic' had the lowest (3.64 to 4.0), while the rest had values between 4 and 5. In both experiments, plants in most cultivars had similar high visual scores of above 4.5 (highest score 5), indicating that all cultivars are suitable for container production in a greenhouse.

### 3.2. Leafy Vegetable Trials in Hydroponic Systems

In the hydroponic systems, all cultivars performed well, but shoot fresh weight (yield) varied among cultivars, as observed in the container culture (Figure 2). Variation among cultivars was generally in agreement with the container culture where 'Beka Santoh', 'Chinese', and 'Ching Chang' had higher yields than the rest of the cultivars. However, in hydroponic culture, there were differences among hydroponic systems due to different planting densities. For cultivars with large size and lateral growth such as Beka Santoh, the NFT system resulted in higher yield because there was more room for the plant to stretch. In addition, plant performance and yield were slightly lower in the second experiment due to high temperatures. The average daily temperature in Expt. 2 was 27 °C, while the daily high temperatures ranged from 30 °C to 33 °C.

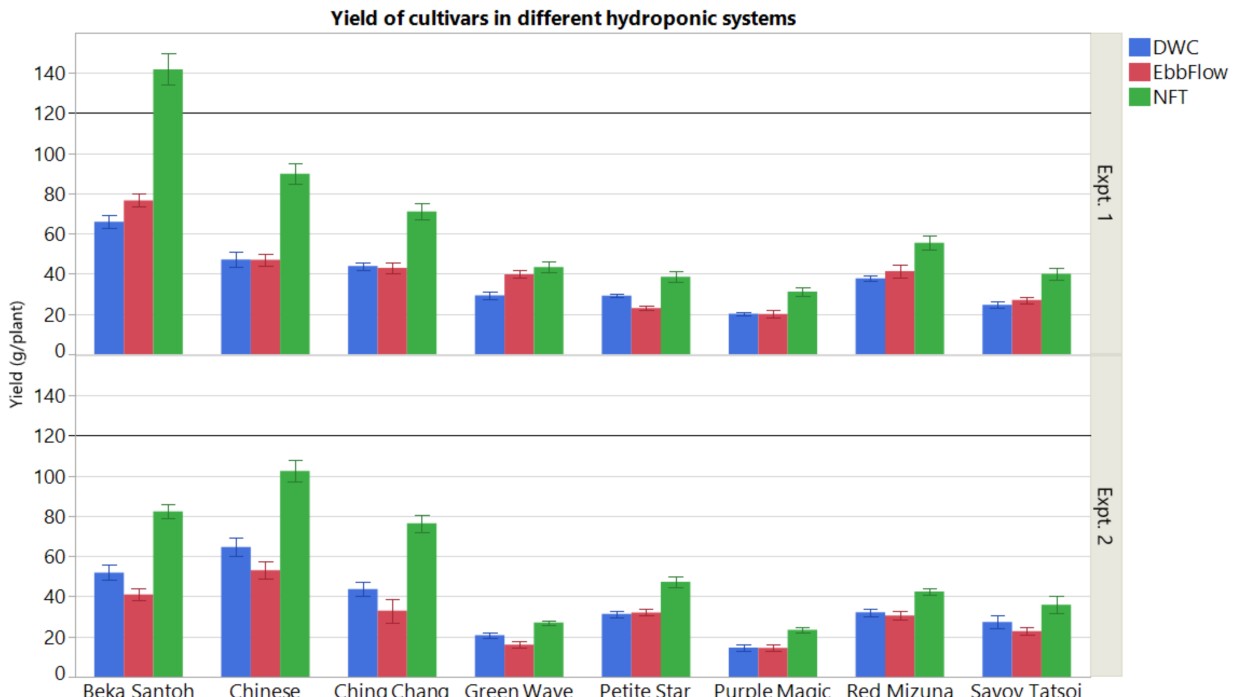

**Figure 2.** Yield of eight leafy vegetables grown in three hydroponic systems: DWC, deep water culture; EbbFlow, Ebb and Flow; NFT, nutrient film technique. Top, Expt. 1 (from 22 April to 23 May 2019); bottom, Expt. 2 (12 June to 3 July 2019).

### 3.3. Leaf Vegetable Trials in the Open Field and High Tunnel in East Texas

Three trials were conducted in east Texas: two in the open field and one in the high tunnel. We evaluated the performance of different cultivars based on the visual scores during the trials, which was based on a scale of 1 (excellent) to 10 (dead), a different scale used in the West Texas trials. For the first planting (Field 1), three weeks after transplant, the visual scores were similarly higher for 'Beka Santoh', 'Chinese', 'Purple Magic', and 'Red Mizuna', while all others had scores above 2.3 (Table 4). In the second rating for the same plants, one week after, 'Chinese' and 'Red Mizuna' had excellent scores numerically, but there were no statistical differences among all cultivars. Plant performance was acceptable. For the second planting of the open field trials (Field 2), due to winter weather conditions, plants did not grow significantly, and plant performance declined, although most of them survived, even though there were days when temperatures reached below freezing. Based on the visual scores, there were no significant differences among cultivars, indicating that all these cultivars were tolerant to cold. All species of Bok choy (*B. rapa*) started to bolt and eventually flowered by early February when the daylength and temperature started to increase. However, the plants in the high tunnel performed better than those in the field, even though there was only a plastic cover over the frame without enclosures of the sides and end doors.

The yield of the nine leafy vegetables in the open field varied with cultivar as observed in greenhouse trials (Table 5). For the first planting (4 October to 16 November 2018), 'Chinese' had the highest yield (180 g/plant), while 'Purple Magic' had the lowest (63 g/plant). The second planting date was too late in the season (6 November) and plants did not grow due to the winter weather with an average low temperature of 3 °C to 4 °C and average daily temperatures around 9 °C. Therefore, yield data were not presented. For the plants in the high tunnel, the yield was higher in 'Beka Santoh', 'Chinese', 'Ching Chang', and 'Petite Star' than those of other cultivars.

For leaf Brix, the highest values were observed in those plants that experienced cold temperatures. Compared to the plants in the high tunnel, the plants in the field after the winter had much higher Brix values. The highest values were observed in cultivars 'Green Wave' and 'Red Mizuna', which are *B. juncea*. There were species differences in leaf Brix.

However, no differences were found in leaf Brix for plants grown in the high tunnel, which were harvested on 14 December.

**Table 4.** Visual quality rating of nine cool-season leaf vegetables grown in the open field with two planting dates and in a high tunnel in east Texas. Rating scale from 1 to 10: 1—excellent, 10—dead. For Field 1, seeds sowing, transplanting, and harvest dates: 17 September, 4 October, and 16 November 2018. For Field 2, seeds sown, transplanting, and harvest dates: 8 October, 6 November 2018, 8 February 2019.

| Cultivar | Field 1 | | | | Field 2 | | | High Tunnel | | |
|---|---|---|---|---|---|---|---|---|---|---|
| | 25 Oct 21 DAT [z] | | 31 Oct 27 DAT | 30 Nov 26 DAT | | 13 Dec 39 DAT | 17 Jan 72 DAT | 30 Nov 27 DAT | 08 Dec 35 DAT | 13 Dec 40 DAT |
| Beka Santoh | 1.7 | B [z] | 2.3 | 2.0 | AB | 2.3 | 3.7 | 2.7 | 1.3 | 2.0 |
| Chinese | 1.7 | B | 1.0 | 3.0 | A | 2.3 | 4.3 | 3.0 | 2.7 | 2.3 |
| Ching Chang | 2.7 | AB | 3.3 | 2.0 | AB | 2.7 | 3.7 | 2.0 | 1.3 | 1.7 |
| Green Wave | 2.3 | AB | 1.7 | 2.3 | AB | 2.0 | 4.0 | 2.0 | 2.0 | 2.0 |
| Petite Star | 4.3 | A | 3.0 | 1.0 | B | 2.0 | 3.0 | 1.3 | 1.0 | 1.3 |
| Purple Magic | 1.7 | B | 2.3 | 2.0 | AB | 2.0 | 3.3 | 2.3 | 2.3 | 2.3 |
| Red Mizuna | 1.7 | B | 1.0 | 1.0 | A | 2.7 | 3.3 | 1.0 | 1.3 | 1.0 |
| Tatsoi Savoy | 3.0 | AB | 3.7 | 1.7 | AB | 2.7 | 3.0 | 1.7 | 1.3 | 2.0 |

[z] Means followed by different letters indicate significant variety differences according to Tukey's honestly significant difference test ($p < 0.05$). Means within columns without any letters indicate no statistical significance among cultivars.

**Table 5.** Yield and Brix of nine cool-season leaf vegetables grown in the open field and a high tunnel in East Texas. The yield of the second field planting is not presented because all the plants in *B. rapa* were bolted and flowered. However, leaf Brix was measured in plants grown both in the high tunnel and the open field (second planting).

| Cultivar | Yield (g/Plant) | | | | Brix (%) | | |
|---|---|---|---|---|---|---|---|
| | Field 1 | | High Tunnel | | Field 2 | | High Tunnel |
| Beka Santoh | 111.1 | AB | 100.4 | A [z] | 4.6 | C | 2.77 |
| Chinese | 180.0 | A | 98.7 | A | 6.6 | BC | 2.40 |
| Ching Chang | 156.1 | AB | 105.8 | A | 5.8 | BC | 2.73 |
| Green Wave | 91.8 | AB | 40.8 | B | 11.4 | A | 4.67 |
| Petite Star | 108.0 | AB | 75.5 | A | 5.4 | C | 3.40 |
| Purple Magic | 63.0 | B | 29.2 | B | 5.8 | BC | 4.00 |
| Red Mizuna | 69.0 | AB | 57.4 | B | 9.1 | AB | 3.70 |
| Tatsoi | 84.8 | AB | - | - | - | - | - |
| Tatsoi Savoy | 72.9 | AB | 53.0 | B | 6.7 | BC | 4.67 |

[z] Means followed by different letters indicate significant variety differences according to Tukey's honestly significant difference test ($p < 0.05$). Means within columns without any letters indicate no statistical significance among cultivars.

Based on the results of both locations, we can conclude that these leafy vegetables can be grown in a greenhouse year-round in Texas. For field production, planting in the early fall and harvest before frost is recommended. Planting in early spring and harvesting before high temperatures arrive should also be possible. Some varieties may bolt easily, thus, early harvesting is important. In fact, some varieties of Bok choy can be sold as bolted shoots, as long as they are still tender.

For the selected leafy vegetables, both container culture and hydroponic culture in the greenhouse are suitable and plants can grow quickly, although yield varied among cultivars. In addition, whitefly and aphids were the main insect pests affecting Bok choy. With soapy water spray, these pest damages can be controlled as shown in our hydroponic studies from spring to summer in 2019. For field production, fall planting is recommended and plants can be harvested when desirable size is reached. Although some varieties of Bok choy can tolerate a temperature as low as −3 °C, there was little growth when temperatures were below 10 °C based on the results in east Texas. Based on our early field trial [19], conducted in west Texas in 2016 with three planting dates (20 September, 5 October, and 20 October), we found that 5 October was the best planting date. This is because, for early

planting (20 September), young seedlings may be susceptible to heat stress in early fall, while transplanting too late resulted in a lower yield or smaller plants before the cold temperatures arrive, which typically occurs in November. In another trial in west Texas, 12 cultivars of Bok choy were successfully grown in an open field in the Fall and were transplanted to the field in late September [20]. To extend the growing season and increase yield, growing these Asian leafy greens in a high tunnel or greenhouse can significantly increase yield and productivity [21–23]. Depending on the desired harvest size, these leafy vegetables may be harvested as early as three weeks after transplanting, thus multiple sequential plantings or harvesting are possible [24].

*3.4. Warm-Season Vegetable Crop Trials in West and East Texas*

For the field trial in west Texas, yield and fruit count were not significantly different among the three cultivars of eggplant and yardlong bean (Table 6, Figures 3 and 4), although 'Yu Long' (yardlong bean) had numerically lower yield and pod count compared to the other two cultivars. 'Akasanjaku' had the most beans with numerically the highest yield but had smaller bean pods. 'Yu Long' produced the least beans with numerically the lowest yield but had the largest bean pods with the greatest fresh weight per pod. For eggplant, the three cultivars had similar yield and fruit count, while fruit shape, color, and appearance differed. No insect pests were identified in the field. In east Texas, mites were observed in mid-season, which required a corrective spray. However, we had five plants per cultivar in 12-L containers filled with potting mix in the greenhouse and all of them had few aphids (data not included in this paper). We had another project using non-Asian eggplant plants in the greenhouse and all plants had severe aphids. We wanted to confirm if the Asian eggplant would have the same problem.

**Table 6.** Yield of warm season crops in West and East Texas.

| Vegetable | Cultivar | Yield (kg/Plant) | | No. of Pods/Fruits | |
|---|---|---|---|---|---|
| | | West Texas (23 April to 8 August 2019) | | | |
| Yardlong Bean | Akasanjaku | 2.45 | | 248.3 | |
| | Kurosanjaku | 2.07 | | 213.6 | |
| | Yu Long | 1.56 | | 125.7 | |
| Eggplant | Millionaire | 8.21 | | 82.6 | |
| | Purple Shine | 8.73 | | 77.6 | |
| | Shoya | 6.34 | | 80.6 | |
| | | East Texas (23 May to 29 July 2019) | | | |
| Yardlong Bean | Akasanjaku | 1.68 | | 154.8 | A |
| | Kurosanjaku | 1.25 | | 128.9 | A |
| | Yu Long | 1.39 | | 85.1 | B |
| Eggplant | Millionaire | 2.85 | A [z] | 22.0 | A |
| | Purple Shine | 1.24 | B | 8.9 | C |
| | Shoya | 1.52 | B | 13.0 | B |

[z] Means followed by different letters indicate significant variety differences according to Tukey's honestly significant difference test ($p < 0.05$). Means within columns without any letters indicate no statistical significance among cultivars.

In the East Texas trial also, no differences were found in the yield of yardlong bean. However, 'Yu Long' had a lower fruit count. For eggplant, the yield of 'Purple Shine' and 'Shoya' was lower than that of 'Millionaire' and the fruit count was lowest in 'Purple Shine', followed by 'Shoya'. The two locations had different results in yield. The higher yield (g/plant) in west Texas was due to the early transplanting date than that in east Texas. Additionally, harvest was terminated when plants were still producing fruits. The purpose of this trial was not to compare the potential yield but rather to examine the suitability of these cultivars for these locations. From the performance in both locations, we concluded that early planting in the spring is important. Similar to tomato and pepper plants, propagating the seedlings in the greenhouse and then transplanting seedlings to the field after the last frost date helps to increase yields [25,26]. This is because it is important

to establish plants with a strong root system to enhance the vegetative growth to achieve high yields before temperatures get too high. When planting too late in the spring, the high temperature in late spring may stress the young plants, which will negatively influence fruit set and, thus, yield.

The suitable range of average monthly temperatures for yardlong bean is between 20 °C to 30 °C [27]. Yardlong bean prefers warm temperature, but high temperature and humid conditions may cause the plants to become susceptible to powdery mildew [28,29]. Coker et al. [30] conducted variety trials of yardlong bean in Mississippi and observed differences in performance among eight varieties. They observed that two cultivars are best suited for the growing conditions in southern Mississippi. However, mosaic viruses may pose a potential production problem.

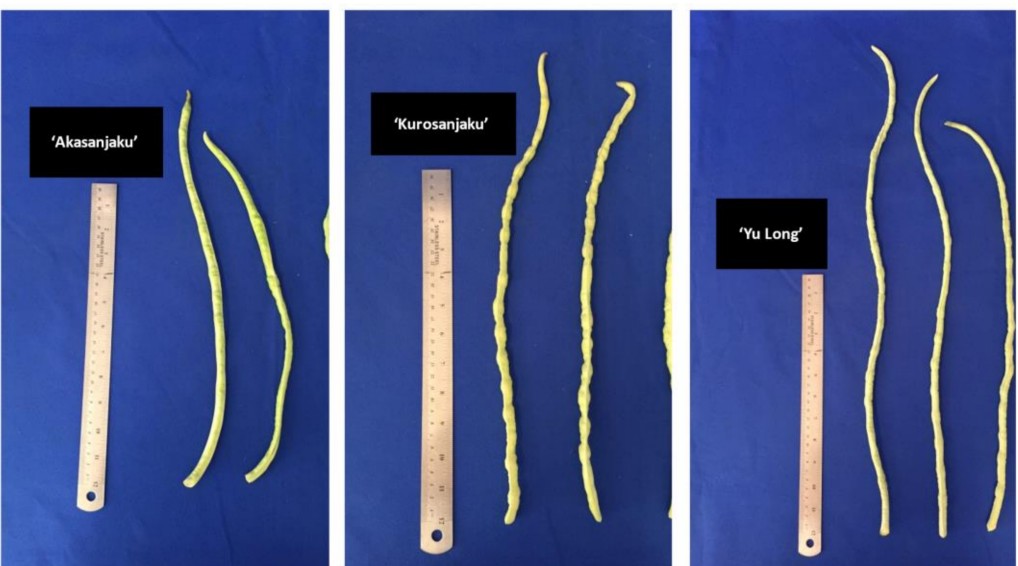

**Figure 3.** Representative photos of the three Yardlong bean cultivars (Akasanjaku, Kurosanjaku, and Yu Long) were harvested from field plots at 8 weeks after transplanting.

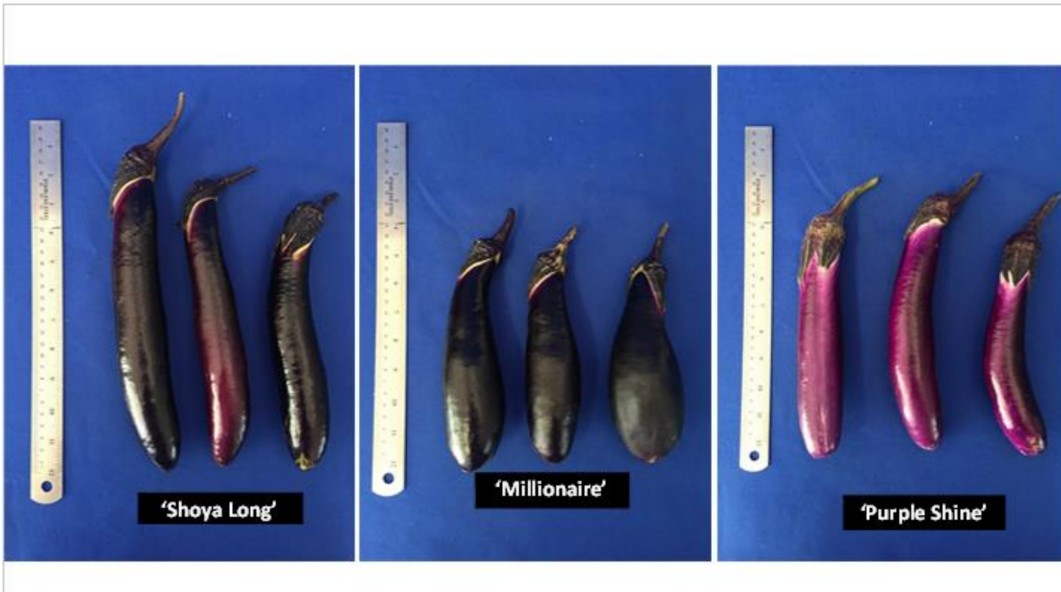

**Figure 4.** Representative photos of the three Chinese eggplant cultivars (Shoya Long, Millionaire, and Purple Shine) harvested from field plots, 6 weeks after transplanting.

## 4. Conclusions and Outlook

Results from two test sites with extreme opposites in climatic and soil conditions indicated a successful production of all tested cool-season and warm-season crops. Insect pressure (aphids, thrips, spider mites) was typical when grown in greenhouses in these two locations. Insects were easily managed with available labeled insecticides. Disease pressure was non-existent for cool-season Asian vegetables as no pesticides were applied in both locations. For the warm season eggplant and yardlong bean varieties, we did not detect pests and diseases in our trials. While data were not presented, eggplant grown in the greenhouse did have aphids but differed on varieties based on visual observation. For all varieties tested, plant yield was influenced by production systems, growing season, and planting density. We concluded that these Asian vegetables have a great potential for Texas climate and regions with similar conditions. More research is needed to test more varieties and growing seasons.

Similar trials of cool-season and warm-season Asian vegetables were also conducted in Uvalde, Texas, and Weslaco, Texas. These diverse locations are representative of the spectrum of environmental conditions (weather, soil types, etc.) and production zones in the Southeast United States. While not presented in tables and figures, data from all these trial locations are consistent with the trends reported in this paper and indicate that production opportunities exist for each region depending on the prevailing resource scenarios. For instance, mild winter conditions in the southernmost region (Weslaco, Lower Rio Grande Valley, Texas), provide a wider production window for cool season vegetables from fall to early spring in the open field. These mild conditions would also permit early planting of some warm-season vegetables such as eggplant and yardlong bean. For more northern locations such as Uvalde (in the wintergarden region, Texas) frost-hardy Asian vegetable varieties would be ideal. Further research is warranted to evaluate disease and pest susceptibility as well as optimal postharvest handling of Asian vegetables in the southeastern United States. As growers continue to search for diversity in their commodity base to increase profits and reduce risk, Asian vegetable production represents an opportunity to diversify while utilizing existing production practices to generate new products for profitable niche markets.

**Author Contributions:** Conceptualization, G.N., J.M., D.L. and J.J.; methodology, G.N., J.M. and T.H.; formal analysis, G.N., T.H., J.M. and G.N.; investigation, T.H., J.M. and G.N.; resources, G.N., J.M., D.L. and J.J.; data curation, J.M. and T.H.; writing—original draft preparation, G.N. and J.M.; writing—review and editing, G.N., J.M., T.H., D.L. and J.J.; supervision, G.N. and J.M.; project administration, G.N. and J.M. All authors have read and agreed to the published version of the manuscript.

**Funding:** This project did not receive any external funding.

**Institutional Review Board Statement:** Not applicable.

**Informed Consent Statement:** Not applicable.

**Data Availability Statement:** Not applicable.

**Acknowledgments:** The authors appreciate all the help received from staff and student workers in all the testing locations.

**Conflicts of Interest:** The authors declare no conflict of interest.

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
