# Peer review of "The Performance of Representative Asian Vegetables in Different Production Systems in Texas"

_agronomy, doi:10.3390/agronomy11091874_

Round 1

Reviewer 1 Report

No coments. The manuscript is very well written.

Author Response

thank you so much for your positive comments. We have revised according to other reviewers' comments and we also improved the English language and style.

Reviewer 2 Report

General comments:

Asian vegetable crops are much more profitable than traditional vegetables and rapidly expanding in the USA, but these “foreign” crops are new to most of American vegetable growers. This manuscript is well written and timely for vegetable growers in Texas and possibly other states as well. As this manuscript may benefit growers the manuscript may fit better to a scholarly journal focusing more on extension.

Specific comments:

  1. The title is too wide and should be concise and informative.
  2. The abstract needs to clearly and precisely (1) state the principal objectives and scope of the investigation; (2) describe the methods used; (3) summarize the results; and (4) tell the principal conclusions. This abstract has missed some important information of the four, particularly, it doesn’t have much data available for readers.
  3. The authors need to give clearer objectives of this study.
  4. What was the soil nutrient and pH backgrounds of the trials?
  5. Why was 15-5-15 used?
  6. Why were no micronutrients applied?
  7. What experimental design was used for the hydroponic trials?
  8. How was the statistical analysis done?
  9. Why the data collected from Uvalde and Weslaco not reported?
  10. The yields of the tested crops look much lower than those of other studies.
  11. After clearer objectives are given, the conclusions should be rewritten and lined up with the objectives.

Author Response

thank you very much for your constructive comments and suggestions. we have revised the manuscript accordingly. Please see the attached PDF file for point-by-point responses.

Reviewer 3 Report

This is an applied experiment report which includes a large number of greenhouse and field trials. Although it has no more theoretical significance, it has certain guiding significance for the producers of Asian vegetables in United States and Europe where the demand is rising rapidly. The writing is fluent and readable. The following suggestions are expected to be supplemented and modified.

  1. L20-23 Add the main finding, results and conclusion in the Abstract.
  2. L135 In order to easy understand the NFT and DWC systems, please add two schematic diagrams how the system works.
  3. L245 Table 3 Delete the unnecessary number after decimal for the fresh weight.
  4. L267 Figure 1 Delete the abscissa description in the figure above, put the two figures together, and share the abscissa description in the figure below. Add the remark “Expt. 1 from 22 April to 278 23 May 2019” in the frame of top figure, and “Expt. 2 from12 June to 03 July 2019” in bottom figure.

Author Response

Thanks so much for your comments and suggestions. The manuscript is revised accordingly. Please see the attached PDF file for point-by-point responses. 

Round 2

Reviewer 2 Report

The authors corrected most of the issues but should also have provided the information on nutrient contents in the soils of the open field trials. This information is needed for readers to understand the results and repeat the trials if they want to.

Author Response

Dear Reviewer,

Thank you so much for your careful and tireless review of our manuscript. I apologize for missing your comment about soil nutrients.  We have added the information in Lines 182 to 187 and Lines 236-237. These are the pre-planting soil test to guide our fertilization management.

Sincerely,

Genhua Niu